# Metabolism of Terephthalic Acid by a Novel Bacterial Consortium Produces Valuable By-Products

**DOI:** 10.3390/microorganisms13092082

**Published:** 2025-09-06

**Authors:** Mitchell Read Slobodian, Dominique Jillings, Aditya Kishor Barot, Jessica Dougherty, Kalpdrum Passi, Sujeenthar Tharmalingam, Vasu D. Appanna

**Affiliations:** 1School of Natural Sciences, Laurentian University, Sudbury, ON P3E 2C6, Canada; mslobodian@laurentian.ca (M.R.S.); sutharmalingam@nosm.ca (S.T.); 2School of Engineering and Computer Science, Laurentian University, Sudbury, ON P3E 2C6, Canadakpassi@laurentian.ca (K.P.); 3Medical Sciences Division, NOSM University, Sudbury, ON P3E 2C6, Canada; 4Health Sciences North Research Institute, Sudbury, ON P3E 2H2, Canada

**Keywords:** PET degradation, TPA metabolism, *Paraburkholderia fungorum*, plastic biodegradation, microbial consortium, benzoate degradation pathway

## Abstract

Terephthalic acid (TPA), a major monomer of polyethylene terephthalate (PET), represents a significant challenge in plastic waste management due to its persistence in the environment. In this study, we report a newly developed bacterial consortium capable of using TPA as the sole carbon source in a defined mineral medium. The consortium achieved stationary phase within five days and metabolized approximately 85% of the available TPA. Metabolite analysis by high-performance liquid chromatography (HPLC) and liquid chromatography tandem mass spectrometry (LC-MS/MS) revealed the activation of the benzoate degradation pathway during TPA catabolism. Additionally, the consortium secreted commercially relevant metabolites such as cis,cis-muconic acid and catechol into the culture medium. Genetic profiling using a reverse transcription quantitative polymerase chain reaction (RT-qPCR) and 16S rRNA sequencing identified *Paraburkholderia fungorum* as the dominant species, suggesting it plays a key role in TPA degradation. The ability of this microbial community to efficiently convert TPA into high-value by-products offers a promising and potentially economically sustainable approach to addressing plastic pollution.

## 1. Introduction

It is widely understood and accepted that plastic pollution produced by humans is an environmental crisis that impacts ecosystems, biodiversity, and the health of living organisms around the globe. When improperly recycled, plastic waste degrades and damages ecosystems by introducing harmful chemicals and plastic derivatives into the environment where they are subsequently consumed by wildlife and disrupt the microbial environment, which all life relies on [1,2,3]. Once in the environment, larger plastic waste decomposes into microplastics (MPs) [4]. There is currently no widely accepted standard for the size of microplastics, but plastic particles 2 mm and less in size are generally considered to be microplastics [4,5]. It is through microplastics that we understand the ubiquity and severity of plastic pollution in our environment and our lives. Not only have microplastics been found in even the most remote environments, but they have also been found inside living organisms, including humans [6,7]. The health effects of microplastic exposure in humans are not yet fully characterized; however, growing evidence indicates that microplastics may worsen existing chronic diseases and potentially contribute to their onset [8,9,10]. Effective solutions to plastic pollution are urgently needed to mitigate its environmental impact, reduce associated greenhouse gas emissions, and protect both ecosystems and human health.

Current methods for plastic disposal have proved to be largely ineffective at preventing and reducing plastic pollution. Broadly, the following four primary strategies are employed to manage plastic waste: recycling, chemical recovery, incineration, and landfills [11]. However, only 11% of global plastic waste is recycled every year while the other 89% is disposed into a landfill, incinerated, or leaked into the environment [11]. Polyethylene terephthalate (PET) is the most commonly used plastic in the world, and only 15.2% of consumer PET is recycled using the current mechanical recycling methods in the U.S. [12]. Furthermore, PET can only be recycled so many times using current mechanical methods as plastics lose a portion of their structural integrity as they are recycled [13]. There are other chemical methods currently in development to increase the recovery rate of polyethylene plastics, such as cleaving ester bonds to depolymerize PET [14]. Current recycling and recovery efforts are focused on degrading PET into monomers, which serve as precursors to valuable by-products in order to help offset the cost [15]. However, the cost of current chemical recycling methods remains a large challenge with regards to widespread adoption [16].

Terephthalate (TPA) and ethylene glycol (EG) are the monomers of PET and each pose a risk to living organisms, particularly via the bioaccumulation of both monomers [17]. PET waste is a global environmental issue, but the capability of PET to bioaccumulate in living organisms is dependent on the particle size [18]. PET waste depolymerizes naturally via natural process like UV radiation and environmental exposure over a long period of time [19]. TPA is an environmental toxicant that bioaccumulates in eukaryotic organisms and causes downstream health consequences [20,21]. To effectively mitigate PET pollution, it is essential to establish a system that can efficiently capture or metabolize the TPA monomer. Therefore, a system that aims to address the issue of PET pollution must also successfully and efficiently remove TPA from the environment.

Recently, microbial systems have been explored as environmentally sustainable and cost-effective alternatives for TPA metabolism and recycling. Many microbial genera have been identified as having the capability to metabolize TPA, including *Acinobacter*, *Achromobacter*, *Arthobacter*, *Bacillus*, *Brevibacillus*, *Cryseobacterium*, *Comamonas*, *Delftia*, *Ideonella*, *Kocuria*, *Microbacterium*, *Pseudomonas*, *Ralstonia*, *Rhodococcus*, *Staphylococcus*, *Stenotrophomonas*, *Streptomyces*, and *Variovorax* [22,23,24,25]. The metabolism of TPA using microbes is a promising area of study, but none have been successfully implemented at a commercial scale and without pre-processing or pre-treatment [26]. The large number and variety of different microbes that have the capability to metabolize TPA presents a unique possibility to find the right organism or community to metabolize plastics effectively and efficiently in a sustainable manner. Microbial systems also provide the advantage of producing valuable by-products like catechol and cis,cis-muconic acid, which can be collected to contribute to the circular economy and help offset the costs of the recycling process [27]. Thus far, the effectiveness of microbes to metabolize TPA has only been successful under modified conditions like higher incubation temperatures, carbon-supplemented media, and chemical pre-treatment of TPA [28]. A microbe or consortium that is capable of efficiently metabolizing TPA at room temperature without the need for carbon supplementation or chemical pre-treatment is needed to effectively combat plastic pollution.

Here, we present a novel consortium of bacteria composed of *Paraburkholderia fungorum*, *Ralstonia pickettii*, *Achromobacter* sp., and *Pseudomonas fluorescens* that is capable of rapidly metabolizing TPA at room temperature without carbon supplementation or chemical pre-treatment while producing valuable by-products. In particular, *P. fungorum* is a novel bacterium in the field of plastic degradation. Previous research has demonstrated its capability to degrade aromatic hydrocarbons, but there has been no research showing its ability to degrade and metabolize TPA to date [29]. Our findings provide new evidence that *P. fungorum*, in combination with a defined bacterial consortium, can metabolize TPA, offering a foundation for future studies aimed at enhancing the efficiency and scalability of microbial plastic degradation.

## 2. Materials and Methods

### 2.1. Liquid Media Preparation, Inoculation, and Collection

Our bacterial consortium was collected from a soil sample obtained from a mine tailing in Sudbury, Ontario. Our bacterial consortium was stored at −80 °C in 50% glycerol stocks. In total, 100 μL of the glycerol stock was used to inoculate 100 mL of liquid culture media, which contained Na_2_HPO_4_ (0.06 g), KH_2_PO_4_ (0.03 g), NH_4_Cl (0.8 g), and MgSO_4_•7H_2_O (0.2 g) per liter of ultrapure water. Next, 1 mL of trace elements (2 μM FeCl_3_•6H_2_O, 1 μM MnCl_2_•4H_2_O, 0.5 μM Zn (NO_3_)_2_•6H_2_O, 1 μM CaCl_2_, 0.25 μM CoSO_4_•7H_2_O, 0.1 μM CuCl_2_•2H_2_O, and 0.1 μM Na_2_MoO_4_•2H_2_O) was added to the medium, then the pH of the medium was adjusted to 6.8 with 2 N NaOH [30]. The carbon source (5 mM TPA or 10% *w*/*v* glycerol) was directly added to 100 mL of the liquid culture. Glycerol was chosen as a control because it is a readily available and easily metabolized carbon source that is commonly used in microbial studies [31,32,33]. Glycerol is a three-carbon aliphatic molecule that bacteria readily metabolize, whereas TPA contains an aromatic ring that necessitates specialized enzymatic machinery for efficient degradation [34,35]. Carbon source concentrations were chosen based on previous work from our laboratory [36,37]. The cultures were incubated in a gyratory shaker at room temperature and at a speed of 120 rpm. The cultures were grown for 7 days, and collections were made roughly every 12 h to monitor cell growth and physiology. Cells were harvested by centrifugation at 11,000× *g* for 20 min at 4 °C. The supernatant was decanted and collected for separate experiments. The cell pellet was washed twice in 1X PBS before being used for downstream experiments. All experiments were performed in biological triplicate.

### 2.2. Cell Physiology Assays and Measurements

The Bradford Protein assay was used to determine the growth of the culture. After washing the cell pellet, the pellet was digested in 100 μL of 1 N NaOH per 1 mL of cell culture collected at 100 °C for 5 min to solubilize the protein content. Bovine serum albumin (BSA) was used as the reference standard, and the experiment was performed in triplicate [38]. The pH of the decanted supernatant fluid was measured to determine the change in pH over time.

### 2.3. 16S rRNA Sequencing

The composition of the bacterial consortium was determined by 16S rRNA sequencing. Cultures grown in TPA were isolated during the exponential growth phase of the culture. DNA was extracted from the cell pellets using a DNA extraction kit (Medi-Res Corp., Sudbury, ON, Canada, Product # G3633-50T). The DNA samples were sent to the McMaster Genomics Facility (McMaster University, Hamilton, ON, Canada) for 16S rRNA amplicon library preparation (V4 region) and sequencing on a MiSeq Illumina instrument (San Diego, CA, USA). DNA samples were prepared and analyzed at the facility using the method previously described by our laboratory [39].

Sequencing data were analyzed using the Illumina BaseSpace 16S Metagenomics application (San Diego, CA, USA), which employs a high-performance implementation of the Ribosomal Database Project (RDP) Classifier as described by Wang Q. et al. [40]. The analysis generated the total number of taxonomic hits per bacterial species for each sample.

### 2.4. LC-MS/MS Analysis

Cultures were grown in either TPA or glycerol (the latter serving as a control), and cells were obtained after 24 h of growth during the exponential growth phase of the culture. Intracellular metabolites were extracted at −20 °C using an extraction solution of 40/40/20 (acetonitrile/methanol/water). The supernatant fluid and extracted metabolites were lyophilized and analyzed via LC-MS/MS (UHPLC: Thermo Scientific Ultimate 3000, Waltham, MA, USA; mass spectrometer: Thermo Scientific Q Exactive; column: Thermo Scientific Hypersil Gold C18, 50 mm× 2.1 mm, and 1.9 μm), as described previously [37]. Metabolite data was analyzed using MetaboAnalyst 6.0 software to determine the differences between the glycerol control and TPA cultures and identify the relevant pathways used to metabolize the TPA. Heatmaps were made to represent the fold change in chemical quantity in the TPA samples compared with the control samples. All chemicals that had a fold change greater than 2 in either the TPA sample or control sample when compared with the system blank were included in the analysis. The heatmap was created using the log of the fold change in order to lower the range of values for increased clarity. Pathway analysis maps were created using *Pseudomonas putida* as the reference species. Pathway analysis graphs depicted the relevance and importance of a pathway on the x-axis and were plotted against the statistical significance between two cohorts on the y-axis [41,42]. A pathway with both high impact and high significance indicated that the pathway was highly relevant in the TPA cultures and not the glycerol control. Metabolites were included if the fold change in quantity when compared with the system blank and the control sample was over 2.

### 2.5. HPLC Analysis

Bacterial cell pellets were sonicated (John’s Scientific Inc., Waltham, MA, USA, model # GE 300) on ice at a power level of 4 [16 s; 5 times with 3 min wait intervals]. In total, 200 μg of cells was added to 2 mL of 2 mM TPA media with or without NaN_3_. Inhibited cells were incubated with NaN_3_ for 30 min prior to the addition of TPA. Samples were taken at times of 0, 30 min, and 2 h. Next, 100 μL of each sample was analyzed using a Waters 2695 HPLC machine (Milford, MA, USA) with a C18 reverse-phase column (Synergi Hydro-RP; 4 μm; 250 mm× 4.6 mm; Phenomenex, Torrance, CA, USA). A Waters Dual Absorbance Detector and Empower software (Empower Pro 2 Software, Build # 2154) were used as described in Alhasawi et al. [43]. The mobile phase was composed of 20 mM KH_2_PO_4_ and 5% (*v*/*v*) acetonitrile at a pH of 2.9, and a flow rate of 0.7 mL/min at an ambient temperature was used to separate the substrates and products.

### 2.6. RT-qPCR Analysis

Bacterial cell pellets were collected, and total RNA was extracted using the Medi-Res RNA Extraction solution (Medi-Res Corp., Sudbury, ON, Canada, Product # G3013-100ML) according to the manufacturer’s instructions. RNA concentration and purity were determined using an ND-1000 NanoDrop spectrophotometer (Wilmington, DE, USA). RNA samples were treated with DNase and reverse-transcribed to complementary DNA using the Medi-Res cDNA Kit (Medi-Res Corp., Sudbury, ON, Canada, Product #: Bi2M-SSRT3). A real-time quantitative polymerase chain reaction (RT-qPCR) was performed using QuantStudio5 (ThermoFisher, Waltham, MA, USA) as previously reported [44]. Primer sequences were designed via Primer-BLAST and validated in-house (Appendix A, Appendix A). The housekeeping genes *rpoA*, *rpoB*, and *groL* were used for normalization. mRNA expression was reported using the ΔΔCT method as a relative mRNA fold increase [45].

## 3. Results

### 3.1. Bacterial Consortium Physiology

The results demonstrate the successful cultivation of a novel bacterial consortium in a liquid medium using TPA as the sole carbon source. The results of the liquid culture physiology are shown in Figure 1. Figure 1A depicts the growth of the bacteria in the liquid media monitored via the measurement of solubilized cellular protein levels. The culture reached the exponential growth phase following a brief lag phase. The exponential growth phase continued until 48 h, when the culture ultimately reached the stationary phase. Figure 1B shows the amount of TPA in the supernatant fluid of the culture over the same time period. TPA consumption mirrored the increase in protein content, suggesting that whilst the bacterial consortium was growing during the exponential phase, the TPA content in the supernatant was being reduced. When the culture reached the stationary phase after 48 h and stopped growing, the TPA content in the supernatant remained consistent. Comparing the two curves suggests that the bacterial consortium was successfully metabolizing TPA in order to grow and stopped consuming the TPA once it reached the stationary phase. The curve in Figure 1C showing the change in supernatant pH follows the same trend as Figure 1B. The pH of the supernatant steadily decreased during the exponential growth phase of the culture until the culture reached the stationary growth phase at 48 h. The decrease in pH was likely due to the production of acidic by-products released from the consortium as a result of TPA metabolism. Taken together, the physiology data presented confirms the overall growth of the consortium and strongly suggests that TPA was being consumed and metabolized by the bacteria while potentially producing valuable by-products in the supernatant fluid, as shown by the decrease in overall pH.

### 3.2. Bacterial Consortium Composition

Our parent bacterial consortium is composed of *P. fungorum*, *R. pickettii*, *Achromobacter* sp., and *P. fluorescens*. Once the capability of the consortium to grow in media containing only TPA as a carbon source was confirmed, 16S rRNA amplicon sequencing was used to determine the change in our consortium composition from the parental line (Figure 2). Interestingly, *P. fungorum* was identified at the species level to comprise 78% of the culture grown in TPA. The majority of the remaining DNA identified belonged to the *Paraburkholderia* genera, although these were likely *P. fungorum* sequences as well, given the large percentage of species-level identifications within our samples. The remaining 2% of the sample comprised the other members of the consortium. Given that the overwhelming majority of the sequences identified were *P. fungorum*, it is likely that the composition of our consortium shifted after being introduced to the TPA carbon source and proves that *P. fungorum* is the major consumer of TPA within our consortium.

### 3.3. TPA Metabolism Is Mediated by the Benzoate Degradation Pathway to Facilitate Oxidative Phosphorylation

To investigate the metabolic fate of TPA following uptake by the bacterial consortium, bacterial pellets were lysed via sonication and the resulting cell-free extracts were analyzed by HPLC. Extracts were either treated with sodium azide (NaN_3_), an electron transport chain (ETC) inhibitor, or left untreated to distinguish between upstream and downstream metabolic intermediates. All conditions revealed the presence of key benzoate degradation intermediates, including cis,cis-muconic acid, catechol, 3-oxoadipyl-CoA, and pyruvate (Figure 3).

Importantly, the uninhibited cell-free extract contained 4-hydroxybenzoate in relatively small amounts, but this was not observed in the inhibited cultures. The presence of 4-hydroxybenzoate in small amounts indicates that TPA was likely being consumed and moved through the benzoate degradation cycle [46,47]. In contrast, the inhibited cells contained a relatively large amount of 3,4-dihydroxybenzoate. Additionally, the quantity of 3,4-dihydroxybenzoate increased with time, suggesting that the inhibition of oxidative phosphorylation by sodium azide slowed the upstream conversion of 3,4-dihydroxybenzoate into relevant TCA cycle metabolites through the benzoate degradation pathway. Collectively, the data support the conclusion that the benzoate degradation pathway plays a central role in TPA metabolism by this bacterial consortium, facilitating the generation of the TCA cycle intermediates necessary for energy production.

### 3.4. Metabolomic Profiling Reveals the TPA-Induced Activation of the Benzoate Degradation Pathway

To assess the metabolic impact of TPA exposure on the bacterial consortium, an untargeted metabolomic analysis of both cell-free extracts and the spent medium from cultures grown with 5 mM TPA as the sole carbon source was performed. A parallel control culture was grown using 10% glycerol to serve as a reference condition for comparison. Samples were collected during the exponential growth phase and assessed via LC-MS/MS analysis. Figure 4 presents heatmaps illustrating the relative abundance of metabolites in TPA-treated cultures relative to glycerol control cultures. In the cell-free extract (Figure 4A), several metabolites involved in the benzoate degradation pathway—specifically 4-hydroxybenzoic acid, benzoic acid, and catechol—were found at significantly higher levels in the TPA condition relative to the glycerol control. Although many metabolites were shared between both conditions, these three formed a distinct cluster in the dendrogram, suggesting coordinated enrichment in response to TPA metabolism. These findings point to the active processing of TPA through aromatic compound degradation pathways [46,47].

The spent-fluid LC-MS/MS analysis (Figure 4B) also showed largely overlapping metabolite profiles between the two culture conditions consistent with the expected background metabolic activity. However, significant increases in catechol and cis,cis-muconic acid were observed in the TPA cultures, suggesting active secretion of these intermediates during aromatic ring breakdown [48,49]. Other research has identified these same metabolites as potential economically viable compounds, but this is the first study to show the production of these compounds by *P. fungorum* [22,23,24,25,27]. The detection of such metabolites in the extracellular medium highlights their potential as valuable by-products of microbial plastic degradation.

Pathway enrichment and impact analyses were then conducted to contextualize the metabolite changes (Figure 5). The most impactful pathway observed in the TPA cell-free extract (Figure 5A) was the benzoate degradation pathway, which metabolizes the TPA derivative 3,4-dihydroxybenzoate to succinyl-CoA. Aminobenzoate degradation was found to be significantly upregulated in the TPA culture, but the pathway had a low impact on cell growth. The high significance but low impact is likely because TPA can be decarboxylated during the degradation of PET to form benzoic acid, a small portion of the larger aminobenzoate pathway, before proceeding to the benzoate degradation pathway [50]. Interestingly, ubiquinone and other terpenoid–quinone biosyntheses had a high impact on both the TPA culture and the glycerol control. In the spent fluid (Figure 5B), the benzoate degradation pathway again showed strong significance and impact, aligning with the extracellular accumulation of catechol and cis,cis-muconic acid. Additional pathways, such as histidine and tyrosine metabolism, were detected in both conditions, but likely reflected core metabolic processes unrelated to TPA catabolism. Taken together, these results confirm that the benzoate degradation pathway is the principal route for TPA metabolism in the bacterial consortium. The secretion of key intermediates like catechol and cis,cis-muconic acid into the growth medium also indicates potential for converting plastic-derived substrates into commercially valuable compounds.

### 3.5. Expression of TPA Degradation Pathway Genes in P. fungorum

Building on our metabolomics data, which identified the benzoate degradation pathway as the main route for TPA catabolism, the expression of key genes involved in TPA metabolism within *Paraburkholderia fungorum* was examined. This species comprised approximately 84% of the consortium population (Figure 2) and is, therefore, presumed to be the primary contributor to TPA degradation. Based on pathway mapping (Figure 6), six genes were selected for RT-qPCR analysis, two from the polycyclic aromatic hydrocarbon degradation pathway responsible for converting TPA into 3,4-dihydroxybenzoate and four from the downstream benzoate degradation pathway. These included genes from both known branches of the pathway that converge at 3-oxoadipate-enol-lactone as well as the gene encoding 3-oxoadipate-enol-lactonase, a key bottleneck enzyme that links aromatic degradation to the TCA cycle. Primers were specifically designed for *P. fungorum* to ensure species-level specificity. Gene expression was measured in both TPA- and 10% glycerol-grown cultures during the exponential phase.

Terephthalate-1,2-dioxygenase had a 200-fold increase in expression in the TPA cultures compared with the glycerol control (Figure 7A). The staggering difference in gene expression was expected because terephthalate-1,2-dioxygenase is the first enzyme in TPA metabolism (Figure 6) and is not required for glycerol metabolism. Interestingly, the other five genes did not have any significant difference in expression compared with the glycerol control, which was unexpected, given that none of these genes are required in glycerol metabolism. To better understand this observation, ΔCt values for each gene were examined (Figure 7B). Interestingly, the five genes showed ample expressions, as demonstrated by Ct values comparable with the reference genes. Therefore, the basal expressions of the downstream genes in the glycerol control were similar to the TPA culture, suggesting that these genes are basally expressed in *P. fungorum* (Figure 7B). The absence of an inducible upregulation may reflect the organism’s inherent readiness to metabolize aromatic intermediates under various conditions. Moreover, the gene expression analysis further established that *P. fungorum* is the primary TPA degrader in the consortium, possessing an innate metabolic capacity for TPA catabolism, and is likely limited by a bottleneck at the initial step catalyzed by terephthalate-1,2-dioxygenase.

## 4. Discussion

The novel bacterial consortium presented in this paper is capable of metabolizing the common consumer plastic derivative TPA as its sole source of carbon while producing valuable by-products. Our consortium mainly comprises the bacterium *P. fungorum*, and is supported by the bacteria *R. pickettii*, *Achromobacter* sp., and *P. fluorescens* (Figure 2). Our consortium grows rapidly in liquid culture media containing only TPA and reaches the stationary phase of its growth by 48 h (Figure 1), all while releasing valuable by-products like cis,cis-muconic acid and catechol into the supernatant fluid (Figure 3 and Figure 4). The consortium first metabolizes TPA through the polycyclic aromatic hydrocarbon pathway to convert TPA into 3,4 dihydroxybenzoate, then continues onto the benzoate degradation pathway where it creates the 3-oxoadipate necessary for the TCA cycle (Figure 5 and Figure 6). Other pathways like the ubiquinone and other terpenoid biosyntheses pathways are also highly impactful for the culture. Ubiquinone biosynthesis is vital for the ETC to function and is a pathway that is expected to be present during the exponential growth phase of any bacterial culture. This could explain the high impact but low difference between the glycerol and TPA cultures, as both should produce ubiquinone. It is possible that the bacteria produce terpenoids that could aid with biological functions like cell permeability to take in the TPA, but it is more plausible that the pathway’s major impact is due to ubiquinone synthesis. Furthermore, a 200-fold increase in terephthalate-1,2-dioxygenase mRNA expression, the first gene responsible for TPA metabolism, clearly demonstrates the active degradation of TPA by *P. fungorum* (Figure 7). Our consortium is clearly capable of innately metabolizing TPA, and *P. fungorum* is the main bacteria responsible for the metabolism of the plastic derivative.

The capability of a wildtype strain of *P. fungorum* to innately metabolize TPA without any genetic modification or physiological assistance is a novel finding in the field of bacterial plastic degradation. The field of bacterial plastic degradation is a relatively well-explored area of research; however, most research has been performed using more typical bacterial models like *E. coli*, *Pseudomonas* sp., *I. sakaiensis*, *Bacillus* sp., and *Comamonas* sp. [11,22,23,24,25,51]. *P. fungorum* is a beneficial soil microbe commonly used in agriculture to aid with nutrient uptake in crops [52,53,54]. *P. fungorum* can also aid in bioremediation as there is some current research demonstrating the capability of *P. fungorum* to metabolize aromatic hydrocarbons [29], but none has looked at plastics or plastic derivatives. Our study provides the first evidence that clearly demonstrates the capability of *P. fungorum* to naturally metabolize plastic derivatives and presents a novel microbe that can be exploited to combat plastic pollution.

The ultimate goal of plastic bioremediation is to naturally degrade these wastes without any chemical or physical pre-treatment. Consumer plastic polymers like PET are physically and chemically stable, making them resistant to natural degradation processes and, therefore, remain in the environment as plastic waste [55,56,57]. PET in particular is a long polymer chain composed of repeating TPA and EG monomers, which need to be metabolized by the bacterium to completely eliminate these forever chemicals [58,59]. EG is a two-carbon aliphatic compound, which is much easier for an organism to break down and metabolize [60,61,62], and preliminary results using our consortium indicate its ability to degrade this dihydroxylated compound. On the other hand, TPA is an aromatic hydrocarbon, which is much more difficult for an organism to convert into a usable source of energy [14]. The aromatic ring in the center of the TPA monomer is incredibly stable and requires specific molecular machinery and enzymes in order to break the ring [14,54,59]. If the organism cannot successfully cleave open the aromatic ring in the TPA monomer after breaking down the PET polymer, then the leftover TPA would still remain as a pollutant. This study has established the baseline for *P. fungorum* degradation of the aromatic monomer of PET. Future work will focus on utilizing the PET polymer itself, as it represents the environmentally relevant form of plastic waste. By initially working with monomers, our study can be certain that our consortium will successfully metabolize plastic waste following the breakdown of the PET polymer.

*P. fungorum* was identified as the primary consumer of TPA within our consortium, and the bacterium metabolizes TPA first through the polycyclic aromatic hydrocarbon pathway followed by the benzoate degradation pathway to create the 3-oxoadipate necessary for the TCA cycle and culture growth (Figure 5 and Figure 6). RT-qPCR analysis of the key genes in both pathways proved that the expression of terephthalate-1,2-dioxygenase was significantly upregulated in the culture containing TPA compared with the glycerol control (Figure 7). Surprisingly, the expression of the key downstream genes involved in the cleavage of the aromatic ring and conversion to a three-carbon aliphatic compound did not change between the TPA and glycerol cultures. The benzoate degradation pathway and these genes are not naturally involved in the metabolism of glycerol [63], so the lack of any difference in expression between the two cultures suggests a few possible conclusions about the metabolism of TPA in *P. fungorum*. First, the significant increase in gene expression for terephthalate-1,2-dioxygenase suggests that *P. fungorum* is the main metabolizer of TPA in the bacterial consortium. Terephthalate-1,2-dioxygenase is the first enzyme in the pathway required to metabolize TPA in bacteria, so it was significantly upregulated compared with the glycerol control where it was relatively sparse. Second, *P. fungorum* is naturally capable of metabolizing TPA. All of the downstream genes involved in TPA metabolism were highly expressed regardless of the carbon source present in the culture, suggesting that the natural basal metabolic expression of the benzoate degradation pathway in *P. fungorum* is sufficient to metabolize the aromatic ring. Working with a bacterium that can innately degrade our plastic of interest is a tremendous advantage when genetically modifying the bacterium in the future. Finally, the lack of any difference in gene expression in the downstream benzoate metabolism genes suggests that *P. fungorum* is innately capable of metabolizing larger amounts of TPA, but the metabolism is currently bottlenecked at the first step by the ability of terephthalate-1,2-dioxygenase to metabolize TPA to TPA-1,2-cis-dihydrodiol. If the basal expression of the downstream TPA metabolism genes is sufficient to metabolize the TPA-1,2-cis-dihydrodiol being produced by the first step in the pathway, then there is likely a bottleneck effect occurring at the first step in TPA metabolism. The bacterium likely increases the expression of terephthalate-1,2-dioxygenase to produce more of the corresponding enzyme and assists in the rapid hydroxylation of TPA, an event that may obviate the need for the increased expression of downstream genes.

We speculate that the upregulation of terephthalate-1,2-dioxygenase may be the result of the upregulation of the entire operon where it is located. *tph2AI* is located downstream of the *Multiple Antibiotic Resistance Regulator* (*MarR*) gene. Proteins within this family respond to stress responses like virulence or the degradation and export of chemicals [64]. The harmful chemicals that activate *MarR* transcription include phenolic compounds, antibiotics, and other aromatic compounds [64,65]. Other soil microbes have been shown to upregulate the *MarR* operon to upregulate downstream genes in order to hydrolyze and metabolize lignin, another polymeric compound made of aromatic moieties [65]. Interestingly, these same bacteria also produce many similar by-products to our consortium, like catechol and 3,4-dihydroxybenzoate, after metabolizing lignin [28]. The similarities between the by-products produced by our consortium and the location of the *tphA2I* gene suggest a shared genetic mechanism for aromatic hydrocarbon degradation amongst these bacteria, but further genetic work is needed to confirm this hypothesis.

The results of our experiments identified multiple by-products being produced by the consortium when consuming TPA. Bacterial plastic consumption is a unique and environmentally friendly option to combat global plastic pollution that achieves economic viability by harvesting the valuable by-products that the bacteria release during plastic consumption [66,67,68]. Some of the compounds identified in the consortium could possibly be extracted from the supernatant of the cell culture for commercial use. Catechol is a TPA derivative and one of the molecules in the benzoate degradation pathway that is metabolized to form 3-oxoadipate, as identified in Figure 5 and Figure 6. Catechol is a compound with a wide variety of uses in multiple different fields and industries, including pharmaceuticals, cosmetics, adhesives, and electroplating [69,70,71]. Cis,cis-muconic acid was also identified in the supernatant of the culture and, like catechol, is also a TPA derivative belonging to the benzoate degradation pathway. Cis,cis-muconic acid can be used to create new plastics and resins [72,73]. Only a fraction of plastics that are recycled by consumers are actually recycled [74,75,76], so another method of plastic recycling would help to increase the percentage of plastic that is actually recycled into new materials instead of going to landfill or becoming plastic pollutants in the environment. Both of these compounds are likely produced in high amounts, given the sharp decrease in pH. Both catechol and cis,cis-muconic acid are acidic compounds, and their release into the supernatant by the bacteria could explain the drop in pH that was observed as the culture grew (Figure 1). Our study did not quantify the production of these metabolites, but future work will focus on quantification and percent extraction from the plastic waste to accurately assess economic viability. Harvesting and isolating these compounds from the supernatant fluid to be sold will be a vital future project towards making the consortium economically beneficial in addition to being a greener solution to plastic waste.

This paper presents a consortium of bacteria that is capable of naturally metabolizing the PET monomer TPA, so the next step for this technology will be to create a culture that can break down the PET polymer itself by inserting the *MHETase* and *PETase* genes into the consortium. MHETase and PETase are enzymes that cleave the PET polymer down to its two subunits, EG and TPA [77,78,79]. First discovered in a strain of *Ideonella sakaiensis* present in a Japanese landfill, the genes for the two enzymes have been inserted and expressed in other bacterial species like *E. coli*, *B. cereus*, and *P. putida* [79]. PETase first cleaves the polymer into units of 2-hydroxyethyl terephthalic acid; the MHETase changes 2-hydroxyethyl terephthalic acid into TPA and EG. Then, the focus should shift to upregulating the expression of the *terephthalate-1,2-dioxygenase* gene to increase the amount of TPA capable of being metabolized or improve the efficiency of the enzyme for a faster conversion of TPA to TPA-1,2-cisdihydrodiol. Leveraging a host such as *P. fungorum*, which exhibits high efficiency in TPA metabolism, could facilitate the development of a microbial system capable of both PET depolymerization and downstream monomer assimilation.

Compared with other monocultures that have had a successful insertion of the *MHETase* and *PETase* genes, our bacterial consortium presents a unique and novel opportunity to successfully upscale the culture from the benchtop to large-scale production. Bacterial consortiums are the gold standard for effective long-term and sustainable bioreactors because bacterial consortiums are resilient and adaptable compared with single strains [80,81]. Although *P. fungorum* is the bacterium largely responsible for the TPA metabolism itself, the other bacteria in the culture (*Achromobacter* sp., *R. pickettii*, and *P. fluorescens*) act as supports in a commensal relationship, resulting in a culture that is more resistant to change [82,83]. Adaptability is vital when degrading consumer plastic waste, which contains numerous additives and different types of plastics that can alter the bacterial environment [84,85]. By using a consortium from the start of our project, we could perform the vital base research required to understand the biology of the consortium and better understand how to upscale the culture to a bioreactor.

Future work on the project will include elucidating the interactions between the bacterial species within the consortium to understand the system as a whole. These goals will be accomplished using a number of different methods. The metabolites being produced in excess that help the community to thrive will be determined. Bacteria exist in a consortium because they live in a commensal relationship, often by producing and releasing vital metabolites required by the other bacteria and vice versa [86,87]. Although *P. fungorum* comprises over 80% of the culture and is the primary metabolizer of TPA, it is likely that there are other metabolites being produced by the other bacteria that are necessary for *P. fungorum* to metabolize the TPA and grow. The ability of the consortium to degrade mixtures of TPA and EG as well as less-processed polymer substrates will also be evaluated. Preliminary experiments indicate that the consortium exhibits enhanced growth in cultures containing a mixture of TPA and EG, highlighting its potential for future applications. Finally, bacteria within our consortium will be isolated and sequenced to fully understand the genetic background of our potentially unique strains. By understanding the broader biology of our consortium, we will create a resilient bacterial consortium capable of being upscaled to the volumes and sizes required for economically viable biotechnology.

## 5. Conclusions

Our work presents a consortium of bacteria that is capable of rapidly growing in a liquid culture containing only the PET derivative TPA as a carbon source. The consortium, which is primary composed of the novel bacterium *P. fungorum* and supported by *R. pickettii*, *Achromobacter* sp., and *P. fluorescens*, metabolizes TPA through the polycyclic aromatic hydrocarbon and benzoate degradation pathways. The use of both of these pathways to metabolize TPA was genetically shown with the 200-fold increase in *tphA2I* expression, the first gene responsible for TPA metabolism. Both catechol and cis,cis-muconic acid were identified as by-products of value being produced by the bacteria in excess, demonstrating our consortium as a possible economically viable green technology. Future work will include the insertion of the *MHETase* and *PETase* genes into the consortium and other experiments required to upscale this plastic-degrading biotechnology. Our novel bacterial consortium and the identification of *P. fungorum* as a metabolizer of TPA present unique and seminal findings that have industrial potential.

## Figures and Tables

**Figure 1 microorganisms-13-02082-f001:**
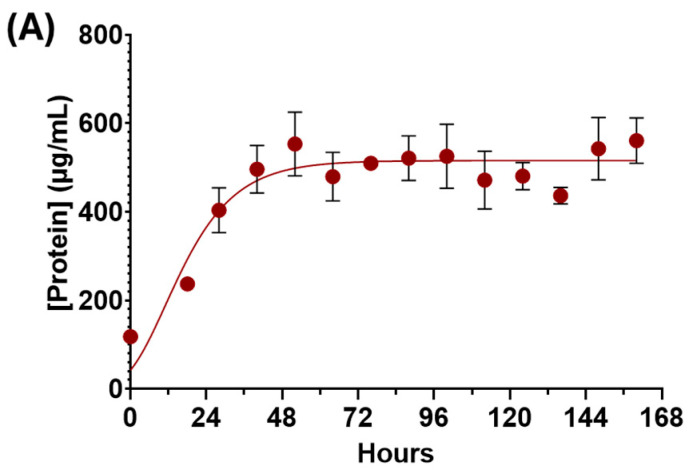
Consortium growth in a 5 mM TPA liquid medium. (**A**) Biomass of the pellet was determined by the Bradford assay. (**B**) Supernatant fluid absorbance was measured directly via UV–vis. (**C**) The pH of the sample was measured by recording the pH of the supernatant fluid. Error bars represent the 95% confidence interval of each sample (*n* = 6).

**Figure 2 microorganisms-13-02082-f002:**
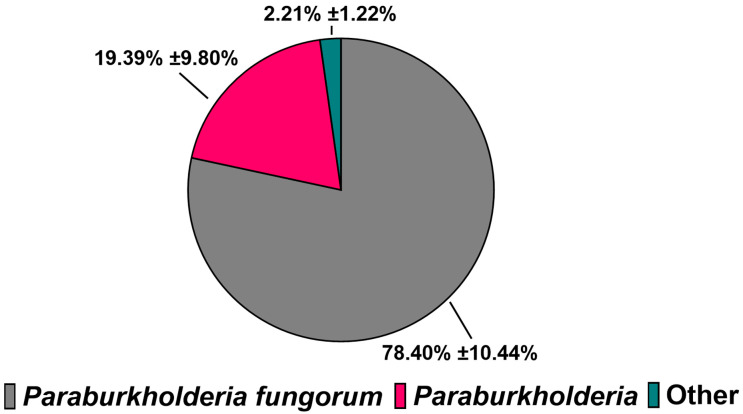
Taxonomic composition of the bacterial consortium following TPA exposure. Relative abundance of bacterial taxa identified in pellet samples from the consortium cultured with TPA. Percentages reflect the proportion of sequences assigned to each taxon. Error values represent the 95% confidence interval (n = 3).

**Figure 3 microorganisms-13-02082-f003:**
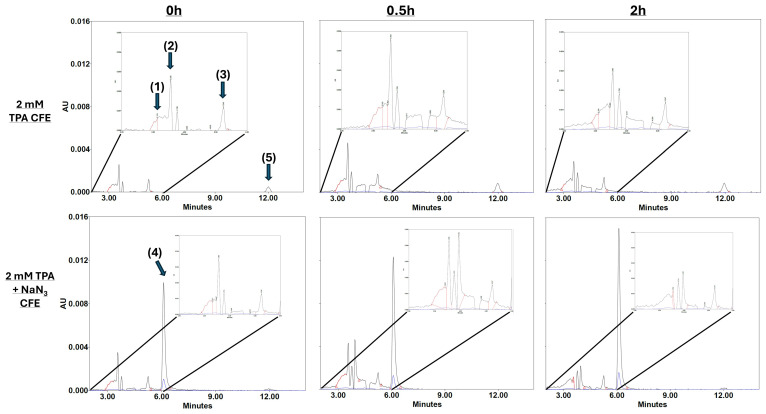
Consortium cell-free extract metabolites in a 2 mM TPA medium reaction buffer measured by HPLC. The compounds identified are as follows: (1) cis,cis-muconic acid, (2) pyruvate, (3) catechol, (4) 3,4-dihydroxybenzoate, and (5) 4-hydroxybenzoate. Arbitrary units (AUs) represent the content of each compound in the samples.

**Figure 4 microorganisms-13-02082-f004:**
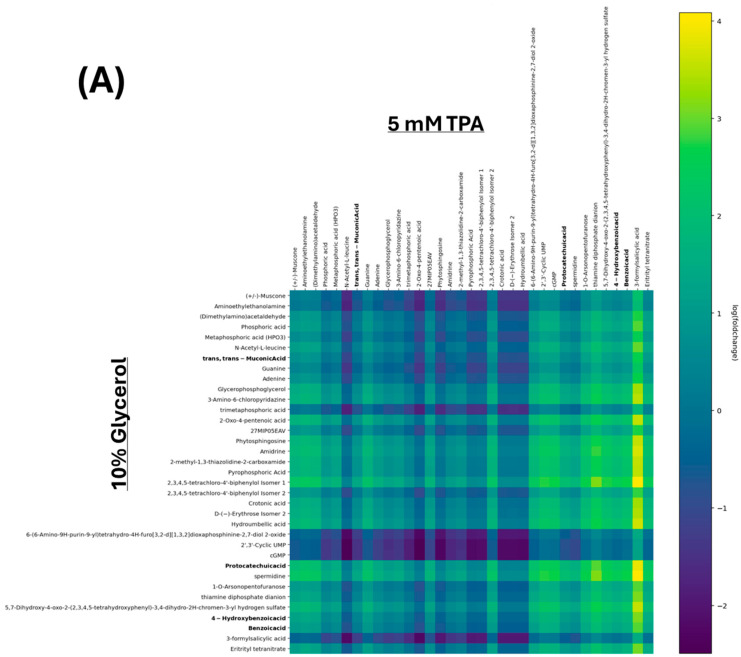
Heatmap comparison of metabolite profiles in TPA- and glycerol-grown bacterial cultures. Heatmaps display relative metabolite abundance in (**A**) cell-free extract and (**B**) spent fluid from cultures grown in 5 mM TPA or 10% glycerol. Lower correlation coefficients indicate greater divergence in metabolite profiles between the TPA and glycerol conditions. Metabolites relevant to plastic degradation and metabolism are highlighted in bold.

**Figure 5 microorganisms-13-02082-f005:**
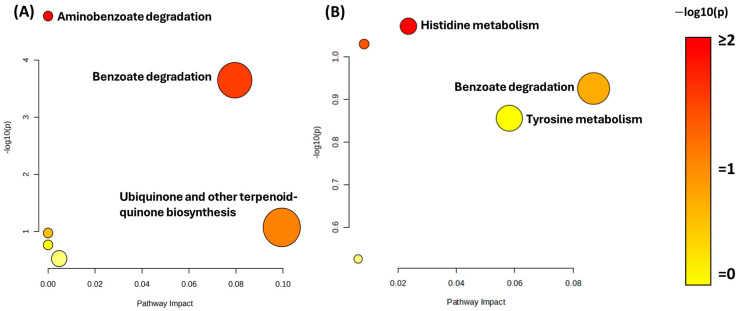
Pathway enrichment and impact analysis of TPA-grown bacterial consortium cultures. Pathway analysis of metabolites identified in (**A**) cell-free extract and (**B**) spent fluid from cultures grown in 5 mM TPA compared with the 10% glycerol control. The y-axis shows statistical significance [−log_10_ (*p*-value)] and the x-axis indicates the pathway impact based on the number and centrality of detected metabolites. Pathways with high significance and impact are most relevant to TPA metabolism.

**Figure 6 microorganisms-13-02082-f006:**
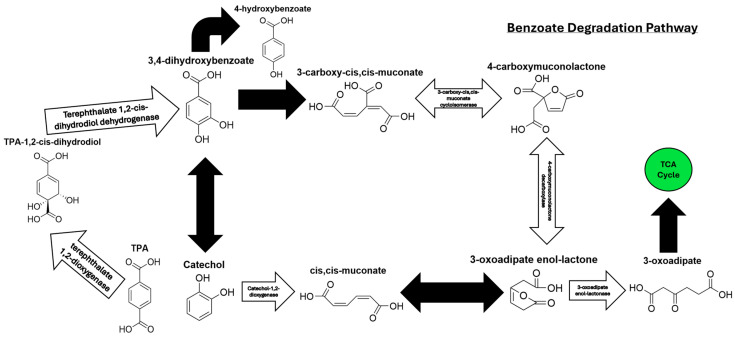
Simplified pathway of TPA metabolism. Simplified version of the polycyclic aromatic hydrocarbon degradation pathway and the benzoate degradation pathway. White arrows are labelled with the enzymes that were targeted for gene expression measurement via RT-qPCR. Black arrows represent metabolites and enzymes present in the samples, but not the focus of the RT-qPCR experiments. The complete pathway is shown in Appendix A.

**Figure 7 microorganisms-13-02082-f007:**
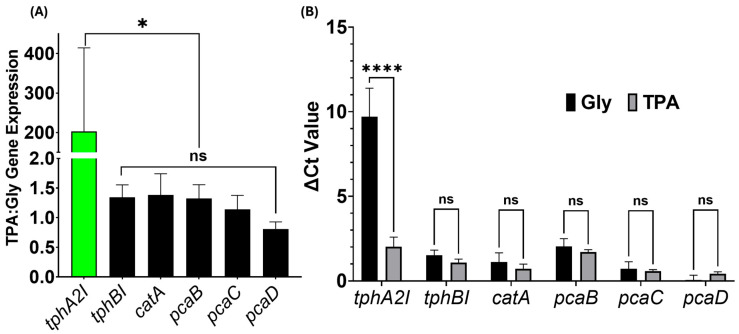
Gene expression of TPA metabolism enzymes in *P. fungorum*. (**A**) Relative mRNA expression of six target genes in TPA-grown cultures compared with the 10% glycerol control during the exponential phase. Green bars indicate significantly upregulated expression in TPA cultures (*** < 0.05, ns = no significance); black bars indicate no significant difference. (**B**) ΔCt values for each gene in TPA and glycerol cultures, normalized to the average Ct of three housekeeping genes. Significance was assessed by a two-way ANOVA (****** < 0.0001, ns = no significance).

## Data Availability

The original contributions presented in this study are included in the article/Appendix A. Further inquiries can be directed to the corresponding author.

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
