# Peer review of "Metabolism of Terephthalic Acid by a Novel Bacterial Consortium Produces Valuable By-Products"

_microorganisms, 2025, doi:10.3390/microorganisms13092082_

Round 1
Reviewer 1 Report
Comments and Suggestions for Authors
The study reports an innovative approach of converting TPA, a precursor of PET, using a novel bacterial inoculum isolated from soil. Overall, the study is relevant as it offers new insights into the biotechnological beneficiation of plastic-based materials through robust microbial cell factories. It also shows us how the tenets of circular bioeconomy could be achieved using microorganisms, which in turn could pave the way for the commercialization of biobased products.
The following comments can improve the scholarly quality of this manuscript:
Abstract:
- Authors must mention the origin of this bacterial strain.
- Which high-value added product was produced or this is a postulation
Introduction:
- The importance of this study needs to be underscored/emphasized, highlighting the commercialization potential of this technological route. Any studies that have been pursued at higher TRL levels.
Methodological approach:
- The approach behind the control experiments is vague, this need to be clearly underscored.
Results:
- Add a section (Table) to compare/contrast your findings with other studies that have explored a similar method using other microbial species, highlighting the benefits/superiority of your results.
Conclusions:
Provide the shortcomings/limitations about your study and propose viable solutions for future studies in this R&D domain.
Reviewer 2 Report
Comments and Suggestions for Authors
The authors of the study present a novel bacterial consortium, dominated by Paraburkholderia fungorum, as an effective decomposer of terephthalic acid (TFA) at room temperature, without the need for any chemical treatments or the addition of extra carbon sources. The study demonstrates the conversion of TFA into intermediates including catechol and cis,cis-muconic acid, through a well-studied benzoate degradation pathway. This work fills several gaps in PET bioremediation, offering economic potential for a fully circular economy. The topic of this research is highly relevant, but some issues require further clarification and minor revisions before the manuscript can be accepted for publication:
- The paper focuses on the degradation of terephthalic acid, a monomer for PET, but it does not address the degradation of PET as a whole. The authors could have given more attention to this important issue, as PET is a major concern in the management of plastic waste.
- When discussing gene expression, it is important to note the relatively low expression of the pcaC and pcaD genes compared to other genes in the TFA degradation pathway.
-
In str. 339, the authors mention that P. fungorum plays a major role in the decomposition of TPA, as it has a strong ability to metabolize this compound. It would be interesting to expand the data on other microorganisms in the consortium, such as Ralstonia pickettii, Achromobacter sp. and Pseudomonas fluorescens, to better understand their roles in this process.
-
The authors emphasize that the microbial community produces valuable intermediates in the culture medium, such as catechol and cis,cis-muconic acid. These metabolites were indeed detected in the medium using HPLC. However, the output of these substances relative to the amount of TFA used at the initial stage has not been provided. Such estimates would be necessary in order to perform an economic analysis of the prospects of the applied TFA biodegradation technology.
Reviewer 3 Report
Comments and Suggestions for Authors
The manuscript ‘’Metabolism of terephthalic acid by a novel bacterial consortium produces valuable by-products’’ addresses an important issue, namely the possibility of PET derivative TPA degradation by a new bacterial consortium that was found to be active at room temperature. This is an advantage because it saves resources needed to heat the system. Metabolite analysis was performed using modern HPLC and LC-MS/MS methods and Paraburkholderia fungorum was identified as the dominant species.
The experimental work performed deserves attention and corresponds to the subject matter of the journal. The text is structured and presented clearly. However, after reading several comments arise.
- The Abstract reflects the study performed and the main results but contains many abbreviations that are not recommended for the Abstract. Please correct.
- Line 126: Please correct.
- Line 144: The brand of the device should be indicated.
- The text contains many personal pronouns that are not welcomed for writing scientific articles (lines 89, 101, 182, 212 etc.) Please correct.
- In Figure 4 the font is illegible. Perhaps the most important points should be bolded, and the font should be enlarged.
